

# Evolutionary history of dimethylsulfoniopropionate (DMSP) demethylation enzyme DmdA in marine bacteria

Laura Hernández[1], Alberto Vicens[2], Luis E. Eguiarte[3], Valeria Souza[3], Valerie De Anda[4] and José M. González[1]

[1] Departamento de Microbiología, Universidad de La Laguna, La Laguna, Spain
[2] Departamento de Bioquímica, Genética e Inmunología, Universidad de Vigo, Vigo, Spain
[3] Departamento de Ecología Evolutiva, Instituto de Ecología, Universidad Nacional Autónoma de México, Mexico D.F., Mexico
[4] Department of Marine Sciences, Marine Science Institute, University of Texas Austin, Port Aransas, TX, USA

Corresponding authors
Laura Hernández,
lhernanj@ull.edu.es
Luis E. Eguiarte, fruns@unam.mx

## ABSTRACT

Dimethylsulfoniopropionate (DMSP), an osmolyte produced by oceanic phytoplankton and bacteria, is primarily degraded by bacteria belonging to the Roseobacter lineage and other marine *Alphaproteobacteria* via DMSP-dependent demethylase A protein (DmdA). To date, the evolutionary history of DmdA gene family is unclear. Some studies indicate a common ancestry between DmdA and GcvT gene families and a co-evolution between Roseobacter and the DMSP-producing-phytoplankton around 250 million years ago (Mya). In this work, we analyzed the evolution of DmdA under three possible evolutionary scenarios: (1) a recent common ancestor of DmdA and GcvT, (2) a coevolution between Roseobacter and the DMSP-producing-phytoplankton, and (3) an enzymatic adaptation for utilizing DMSP in marine bacteria prior to Roseobacter origin. Our analyses indicate that DmdA is a new gene family originated from GcvT genes by duplication and functional divergence driven by positive selection before a coevolution between Roseobacter and phytoplankton. Our data suggest that Roseobacter acquired *dmdA* by horizontal gene transfer prior to an environment with higher DMSP. Here, we propose that the ancestor that carried the DMSP demethylation pathway genes evolved in the Archean, and was exposed to a higher concentration of DMSP in a sulfur-rich atmosphere and anoxic ocean, compared to recent Roseobacter eco-orthologs (orthologs performing the same function under different conditions), which should be adapted to lower concentrations of DMSP.

## INTRODUCTION

Dimethylsulfoniopropionate (DMSP) is an osmolyte synthesized by oceanic phytoplankton and bacteria (*Galinski, 1995*; *Yoch, 2002*; *Curson et al., 2017*). This molecule became abundant in the oceans approximately 250 million years ago (Mya), coinciding

with the expansion and diversification of dinoflagellates (*Bullock, Luo & Whitman, 2017*). Since then, it has played an important role in the biogeochemistry of sulfur cycle on Earth (*Lovelock, 1983*). DMSP is the main precursor of the climate-relevant gas dimethylsulfide (DMS; *Reisch, Moran & Whitman, 2011*). In marine ecosystems, DMSP is rapidly degraded by different bacterial communities (*González, Kiene & Moran, 1999*), and some strains seem to be very efficient and even become dependent on its degradation (*Tripp et al., 2008*). In fact, DMSP supports up to 13% of the bacterial carbon demand in surface waters, making it one of the most significant substrates for bacterioplankton (*Kiene et al., 1999*; *González, Kiene & Moran, 1999*). *Candidatus* Pelagibacter ubique (SAR11), dominant in the bacterioplankton and especially in surface waters, can only use sulfur atoms derived organic molecules, such as DMSP (*Tripp et al., 2008*). In the case of *Ruegeria pomeroyi* DSS-3, a model organism for DMSP studies, the turnover rate of DMSP transformation depends on salinity conditions (*Salgado et al., 2014*).

The first step in the degradation of DMSP involves two competing pathways, cleavage and demethylation. The DMSP cleavage pathway metabolizes DMSP with the release of DMS (*Kiene et al., 1999*), a step catalyzed by a number of enzymes (*Curson et al., 2011*). In the alternative pathway, DMSP is first demethylated by a DMSP-dependent demethylase A protein (DmdA; *Howard et al., 2006*). Compared to genes in the DMS-releasing pathway, *dmdA* is more frequently found in the genomes of oceanic bacteria (*Newton et al., 2010*; *Todd et al., 2009*). The DmdA enzyme was originally annotated as a glycine cleavage T-protein (GcvT) in the model bacteria *R. pomeroyi* (*Reisch, Moran & Whitman, 2011*), although it forms a separate clade from the known GcvTs (gcvT and Unchar. AMT) (*Sun et al., 2011*; *Bullock, Luo & Whitman, 2017*). Despite their structural similarity which might indicate a common ancestry, DmdA and GcvT are mechanistically distinct (*Schuller et al., 2012*). DmdA produces 5-methyl-THF from DMSP as the result of a redox-neutral methyl transfer, while GcvT produces glycine to 5,10-methylene-THF from glycine (*Reisch, Moran & Whitman, 2008*).

Nearly all known DMSP-catabolizing bacteria belong to the phylum *Proteobacteria* with DmdA orthologs found in most of the sequenced members of the *Rhodobacteraceae* family, as well as bacterioplankton strains of SAR11, SAR324, SAR116 and in marine *Gammaproteobacteria* (*González, Kiene & Moran, 1999*; *González, 2003*; *Howard et al., 2006*; *Bürgmann et al., 2007*; *Reisch, Moran & Whitman, 2008*) like Chromatiales which could have gotten DmdA gene by Horizontal gene transfer (HGT) as some studies suggest (*Howard et al., 2006*; *González et al., 2019* ). This phylogenetic distribution suggests an expansion of *dmdA* through HGT events between different lineages of bacteria, presumably through viruses (*Raina et al., 2010*). Since an episode of genome expansion of Roseobacter, predicted early in its genome evolution, coincides with the diversification of the dinoflagellates and coccolithophores around 250 Mya (*Luo et al., 2013*; *Luo & Moran, 2014*), it has been suggested a co-evolutionary event between Roseobacter and the DMSP-producing-phytoplankton (*Luo et al., 2013*; *Luo & Moran, 2014*; *Bullock, Luo & Whitman, 2017*). Under this scenario, the enzymes of the DMSP demethylation pathway could have evolved within the last 250 Mya, as phytoplankton responded to the marine catastrophe at the end of the Permian, with the diversification of dinoflagellates that

produce DMSP and the Roseobacter clade expanding by using DMSP as its main sulfur source. Despite this hypothesis, there is a lack of knowledge about the main evolutionary events that lead the adaptation to DMSP in Roseobacter.

The biosynthesis of DMSP has been reported in marine heterotrophic bacteria, such as the *Alphaproteobacteria*, that is, *Labrenzia aggregata* (*Curson et al., 2017*), *Gammaproteobacteria* and *Actinobacteria* (*Williams et al., 2019*). Moreover, bacteria seem to be important producers of DMSP and DMS in coastal and marine sediments (*Williams et al., 2019*). Since the common ancestor of heterotrophic bacteria and Roseobacter originated in the Archean, more than 2 billion years ago (*Kumar et al., 2017*), the Roseobacter and other *Alphaproteobacteria* might have been exposed to DMSP early (*Reisch, Moran & Whitman, 2011*; *Reisch et al., 2011*). According to this hypothesis, the DMSP demethylation and the cleavage pathways arose by the evolution of enzymes that were already present in bacterial genomes and adapted in response to the wide availability of DMSP. As mentioned earlier, *Alphaproteobacteria* in the SAR11 group seems to thrive at the expense of organic sulfur compounds, such as DMSP, and had a common ancestor that lived ca. 826 Mya, at the end of the Precambrian (*Luo et al., 2013*). We would then expect a common ancestor of the DmdA gene family during the early Proterozoic and that the functional divergence between DmdA and GcvT gene families was driven by both functional constraints and widespread HGT, probably during the Huronian snowball Earth, a period of planetary crisis where the greatest microbial diversity took refuge in the shallow seas close to the equator (*Tang, Thomas & Xia, 2018*).

Here, we analyzed the evolutionary history of the DmdA gene family in marine *Proteobacteria* by considering three evolutionary scenarios: (1) a recent common ancestry of DmdA and GcvT, (2) a coevolution between Roseobacter and the DMSP-producing-phytoplankton, and (3) an enzymatic adaptation for utilizing DMSP in marine bacteria prior to Roseobacter origin. We first analyzed if convergent, independent or HGT-based evolution can explain the presence of *dmdA* genes in different bacterial lineages SAR11, SAR116 and *Rhodobacteraceae*. Then, we inferred the most recent common ancestor (MRCA) of the DmdA gene family, the timing of its origin and any duplication events. We also reconstructed the ancestral forms of DmdA enzymes to infer the most likely ecological conditions where DmdA thrive. We provide insights into their function by analyzing DmdA structural evolution. Finally, we examined how natural selection could have driven the divergence of the DmdA gene family. Our results indicate that *dmdA* appeared before the origin of the Roseobacter clade and the conditions of the late Permian created by eukaryotic phytoplankton. Therefore, DmdA is an adapted version of enzyme that evolved in response to the availability of DMSP.

## MATERIALS AND METHODS

### Data mining

Peptides and genes from DmdA gene family were collected from a set of 771 genomes manually curated and hosted in the MarRef database (*Klemetsen et al., 2018*). The DmdA orthologs and homologs sequences were obtained as described by *González et al. (2019)*. The DmdA homologs included were obtained using a HMM designed for DmdA

orthologs (*González et al., 2019*), with a relaxed maximum *e*-value (E−50). A total of 204 sequences from 184 genomes were used to infer the evolutionary history of DmdA gene family (Table S1).

## Phylogenetic tree reconstruction and topology tests

The phylogenetic tree of the DmdA protein sequences included DmdA orthologs and DmdA homologs (non-DmdA as in *González et al. (2019)*). The sequences were aligned using MUSCLE (*Clamp et al., 2004*; *Edgar, 2004*). Regions poorly aligned or with gaps were removed using TrimAl (*Capella-Gutiérrez, Silla-Martínez & Gabaldon, 2009*) with parameters set to a minimum overlap of 0.55 and a percent of good positions to 60. Best-fit evolutionary model was selected based on the results of the package ProtTest 3 (*Darriba et al., 2011*) to determine the best-fit model for maximum likelihood (ML) and Bayesian inference (BI).

For the maximum likelihood analysis (ML), PhyML v3.0 (*Guindon et al., 2010*) or RAxML v7.2.6 (*Stamatakis, 2006*) were used to generate 100 ML bootstrap trees, using the Le Gascuel (LG; *Le & Gascuel, 2008*) model with a discrete gamma distribution (+G) with four rate categories, as this was the model with the lowest Akaike information criterion and Bayesian information criterion score. For the Bayesian analysis (BI), trees were constructed using the PhyloBayes program (*Lartillot & Philippe, 2004*, *2006*; *Lartillot, Brinkmann & Philippe, 2007*) with the CAT model that integrates heterogeneity of amino acid composition across sites of a protein alignment. In this case, two chains were run in parallel and checked for convergence using the tracecomp and bpcomp scripts provided in PhyloBayes. As an alternative, we computed a phylogenetic tree using a BI implemented in BEAST2 program which was run with relaxed clock model and Birth Death tree prior (*Bouckaert et al., 2014*). Finally, we used R v3.6.1 (*R Core Team, 2017*) with phangorn v2.5.5 (*Schliep, 2011*) to perform consensus unrooted trees.

We ran several topology tests to establish whether the trees generated using the ML and BI methods provided an equivalent explanation for the two main groups, that is, the non-DmdA and DmdA clades. For this analysis, the topologies were compared with the TOPD/FMTS software v4.6 (*Puigbo, García-Vallve & McInerney, 2007*). A random average split distance of 100 trees was also created to check if the differences observed were more likely to have been generated by chance.

## HGT test and GC content analysis

Two approaches were used to detect HGT. First, a phylogenetic incongruence analysis (*Ravenhall et al., 2015*) through three topology tests, the Kishino-Hasegawa (KH) (*Kishino & Hasegawa, 1989*), the Shimodaira-Hasewaga (SH) (*Shimodaira & Hasegawa, 1999*) and the approximately unbiased (AU) (*Shimodaira, 2002*), implemented in the IQ-TREE software v1.5.5 (*Nguyen et al., 2015*). Two topologies were tested, the ML topology obtained for the species tree of the genomes here analyzed, and the ML phylogeny of DmdA. To construct the species tree, ribosomal protein 16 small subunit (RPS16) sequences were collected from the MarRef database (*Klemetsen et al., 2018*), one for each genome (Table S1).

The GC content variation was studied to identify genes that have a different percentage of GC content at the third position of codons with respect to the neighboring genomic regions. The EPIC-CoGe browser (*Nelson et al., 2018*) was used to visualize the genomes and sequences and look for genes that use different codons with respect to the rest of the genomic dataset (data are available under permission as "ULL-microevolution" on https://genomevolution.org/).

## Molecular dating

We first tested for heterogeneities in the substitution rates of the genes using a likelihood ratio test (LRT) (*Felsenstein, 1981*) with the ML-inferred tree. Likelihoods' values were estimated using baseml in PAML v4.8 (*Yang, 2007*) under rate constant and rate variable models and used to compute the likelihood ratio test (LRT) statistic according to the following equation:

$$LRT = -2(\log L_1 - \log L_0)$$

where $L_1$ is the unconstrained (nonclock) likelihood value, and $L_0$ is the likelihood value obtained under the rate constancy assumption. LRT is distributed approximately as a chi-square random variable with (m-2) degrees of freedom (df), m being the number of branches/parameters.

To conduct a molecular dating analysis with BEAST 2 (*Bouckaert et al., 2014*), two independent MCMC tree searches were run for 50 million generations, with a sampling frequency of 1,000 generations over codon alignment obtained, as we explain in the next section. The GTR substitution model with a gamma shape parameter and a proportion of invariants (GTR + G + I), was selected with PartitionFinder software v2.1.1 (*Lanfear et al., 2016*) based on the Bayesian Information Criterion (*Darriba et al., 2012*), applied with a Birth Death tree prior (*Gernhard, 2008*) and an uncorrelated relaxed clock log-normal. The molecular clock was calibrated using information from the TimeTree database (*Hedges, Dudley & Kumar, 2006*; *Hedges et al., 2015*; *Kumar et al., 2017*). We used the proposed dates of the MRCA of (1) the *Alpha-* and *Gammaproteobacteria* (2,480 Mya), (2) the *Halobacteriales* (455 Mya) (Figs. S1–S3) (*Hedges, Dudley & Kumar, 2006*; *Hedges et al., 2015*; *Kumar et al., 2017*), and (3) the SAR11 (826 Mya) (*Luo et al., 2013*). A log-normal prior distribution on the calibrated nodes centered at the values mentioned above was specified with 20 standard deviations and constrained to be monophyletic. Convergence of the stationary distribution was checked by visual inspection of plotted posterior estimates in Tracer v1.6 (*Rambaut & Drummond, 2013*) to ensure effective sample sizes (ESSs) of parameters were >>200, as recommended by the authors. After discarding the first 15% trees as burn-in, the samples were summarized in the maximum clade credibility tree using TreeAnnotator v1.6.1 (*Rambaut & Drummond, 2002*) with a PP limit of 0.5 and summarizing mean node heights. Means and 95% higher posterior densities (HPDs) of age estimates are obtained from the combined outputs using Tracer v1.6. The results were visualized using FigTree v.1.4.3 (*Rambaut, 2009*).

## Maximum likelihood tests of positive selection

To measure the strength and mode of natural selection during the evolution of DmdA gene family, the ratio of non-synonymous (dN) to synonymous substitutions (dS) ($\omega$ = dN/dS) was calculated in CodeML implemented in the suite Phylogenetic Analysis by Maximum Likelihood (PAML package v4.8) (*Yang, 2007*).

CodeML requires an alignment of coding sequences, and a phylogenetic tree. DNA alignment was achieved by MUSCLE (*Edgar, 2004*) implemented in MEGA-CC v7.0.26 (*Kumar, Stecher & Tamura, 2016*) and poorly aligned segments were eliminated with Gblocks under defaults parameters (*Castresana, 2000*). The phylogenetic tree was built using ML with PhyML v3.0 (*Guindon et al., 2010*) as described above and a nucleotide substitution model selected by jModelTest (*Darriba et al., 2012*). DAMBE (*Xia, 2001*) was also used to check for saturation of nucleotide substitutions using a plot of the number of transitions and transversions for each pairwise comparison against the genetic distance calculated with the F84 model of nucleotide substitution (*Huelsenbeck & Rannala, 1997*), which allows different equilibrium nucleotide frequencies and a transition rate-transversion rate bias. Multiple sequence alignments with similar characteristics (i.e., showing saturation of nucleotide substitutions) were then analyzed with CodeML (*Yang, 2007*).

Three sets of models were used (site-specific, branch-specific and branch-site models) to detect pervasive and episodic selection during the evolution of *dmdA* orthologs. Likelihood-ratio tests (LRTs) were used to compare models, and significant results (*p*-value < 0.05) were determined contrasting with a chi-square distribution (chisq) (*Anisimova, Bielawski & Yang, 2001*).

In the site-specific analysis, we tested for variability of selection (type and magnitude) across the codons of the gene using three pairs of nested models. The first pair includes M0 (just one dN/dS ratio) and M3 ("K" discrete categories of dN/dS) and has four degrees of freedom (df). The second pair of models considers M1a (just two classes of sites, purifying (dN/dS < 1) and neutral selection (dN/dS = 1)) and M2a (the same as M1a adding a third class of sites dedicated to positive selection (dN/dS > 1)), this has two df. Finally, the third pair of models comprised M7 (a beta distribution that allows dN/dS to vary among the interval (0, 1)) and M8 (adds an extra discrete category to M7 with dN/dS > 1), with two df. Whereas M0 vs M3 tests for evidence of dN/dS variation across sites, M1a vs M2a and M7 vs M8 tests for the presence of sites under positive selection (dN/dS > 1).

Using three branch models (*Yang, 1998*), we tested for variation of selection over evolutionary time. The null model (M0) assumes that all branches evolve at the same rate, therefore, there is only one value of dN/dS for all the branches of the tree. The two-ratio model allows two dN/dS values, one value for all the Roseobacter lineage (we called this group A) and another for the rest of branches (group B). The free-ratio model, allows one dN/dS value for each branch. Null and two-ratio model are compared by LRT with one df but null and free-ratio model are compared with 36 df.

For the last set of models, we identified sites that have been under positive selection at a particular point of evolution using branch-site models, in which dN/dS can vary among sites and among branches (*Yang & Dos Reis, 2011*; *Zhang, 2005b*). We computed two models: a null model, in which the "foreground branch" may have different proportions of sites under neutral selection to the "background branches", and an alternative model in which the "foreground branch" may have a proportion of sites under positive selection. We compare these models for each terminal branch with a LRT of one df. For each branch-site analysis, we applied the Bonferroni correction for multiple testing.

In site and branch-site tests, we identified sites under positive selection as those with Bayes Empirical Bayes (BEB) posterior probability above 0.95 (*Yang, 2005*). We also checked for convergence of the parameter estimates in PAML by carrying out at least two runs for each tree and starting the analysis with different ω (0.2, 1, 1.2 and 2). In addition, to test for convergent selection in several lineages, we ran at branch-site analysis selecting as "foreground branches" all those under positive selection in a previous analysis.

## Analysis of functional divergence

Divergent selection is indicated by different ω values among paralogous clades. We tested whether selective pressures diverged following duplication that led to *dmdA* and non-*dmdA* genes (*Bielawski & Yang, 2004*). We compared the M3 model, which accounts for ω variation among sites but not among branches or clades, with a model allowing a fraction of sites to have different ω between two clades of a phylogeny (clade model D). We also tested M0 and M3 models and we used a posterior BEB probability above 0.95 to identify sites evolving under divergent selective pressures. We checked for convergence of the parameter estimates in PAML by carrying out at least two runs for the tree and starting the analysis with different ω (0.1, 0.25, 2, 3 and 4).

Finally, we applied two branch-site models (as described above) to test dN/dS differences on the branches representing the ancestral lineages of the DmdA and non-DmdA clades (see "Results"). We considered the ancestral sequences from DmdA and non-DmdA clades as foreground branches in two different models.

## Reconstruction of ancestral DmdA sequence

To reconstruct the ancient conditions where *dmdA* gene prospered, we inferred the ancestral sequences of the DmdA node using the FastML web server (*Ashkenazy et al., 2012*) and then computed estimated physico-chemical properties on predecessor sequence using Compute ProtParam tool from Expasy—SIB Bioinformatics Resource Portal (*Gasteiger et al., 2005*). Moreover, we also reconstructed the ancestral sequence of the non-DmdA node, as well as the ancestral sequence of both the DmdA, and the non-DmdA families. FastML was run considering the alignment of proteins and the ML phylogenetic tree for those DmdA orthologs or homologs inferred as we explained above. Posterior amino acid probabilities at each site were calculated using the LG matrix and Gamma distribution. Both marginal and joint probability reconstructions were performed. Protein

sequences resulting from marginal reconstructions were used to predict tertiary structure (see below) as well as to identify family domains using Pfam v32 (*Finn et al., 2010*).

## Protein tertiary structure analysis

Predicted three-dimensional structures of protein sequences were examined by Iterative Threading ASSEmbly Refinement (I-TASSER) (*Roy, Kucukural & Zhang, 2010*; *Yang et al., 2015*). First, I-TASSER uses local meta-threading-server (*Wu & Zhang, 2007*) to identify templates for the query sequence in a non-redundant Protein Data Bank (PDB) structure library. Then, the top-ranked template hits obtained are selected for the 3D model simulations. To evaluate positively the global accuracy of the predicted model, a *C*-score should return between −5 and 2. At the end, the top 10 structural analogs of the predicted model close to the target in the PDB (*Berman et al., 2000*) are generated using TM-align (*Zhang, 2005a*). The TM-score value scales the structural similarity between two proteins and should return 1 if a perfect match between two structures is found. A TM-score value higher than 0.5 suggests that the proteins belong to the same fold family.

We used PyMol v1.7.4 (*DeLano, 2002*) to visualize the 3D structure of the proteins and to map the positively selected sites onto the 3D structure of DmdA (pdb: 3tfh).

# RESULTS

## Phylogenetic tree for DmdA family

We identified a total of 204 DmdA protein sequences out of 150 curated genomes (see Table S1: Genomes and genomic diversity sheets), and reconstructed their evolutionary relationships using BI (Fig. 1) and ML (Fig. S4). Unrooted trees in TOPD-FMTS showed that split distances did not exceed 0.19, indicating that the phylogenetic reconstruction is robust, with minor variations in alignment filtering and methods for inferring topologies (Table S2).

The BI tree (Fig. 1) shows a main duplication between two lineages. The larger phylogenetic group comprises genes mainly from *Bacteroidetes*, while the smaller group includes genes mainly from *Alphaproteobacteria*. We focused on this smaller group as it includes the DmdA sequences (Fig. 1; green color) and the closest homologs to DmdA (Fig. 1; brown color).

Using phylogenetic analyses including DmdA orthologs and DmdA homologs close to those (the limit to select the closer homologs was set to a maximum *e*-value of E−80), we resolved the position of the first DmdA sequences isolated from two marine bacterial species, *R. pomeroyi* (AAV95190.1) and *Ca.* P. ubique (AAZ21068.1). In addition, the inclusion of DmdA homologs allowed to resolve a robust phylogenetic relationship of the DmdA gene family (Fig. 2). We detected a clear separation between DmdA and putative non-DmdA families. Indeed, the four DmdA family trees constructed using different methods compared in TOPD-FMTS using split distances (Table S3) and unrooted trees (Fig. S5) agreed with this result. The average split distance was 0.60, indicating that the trees were neither identical (split difference = 0) nor completely different (1). A random split distance was calculated to analyze whether the split distances were significantly

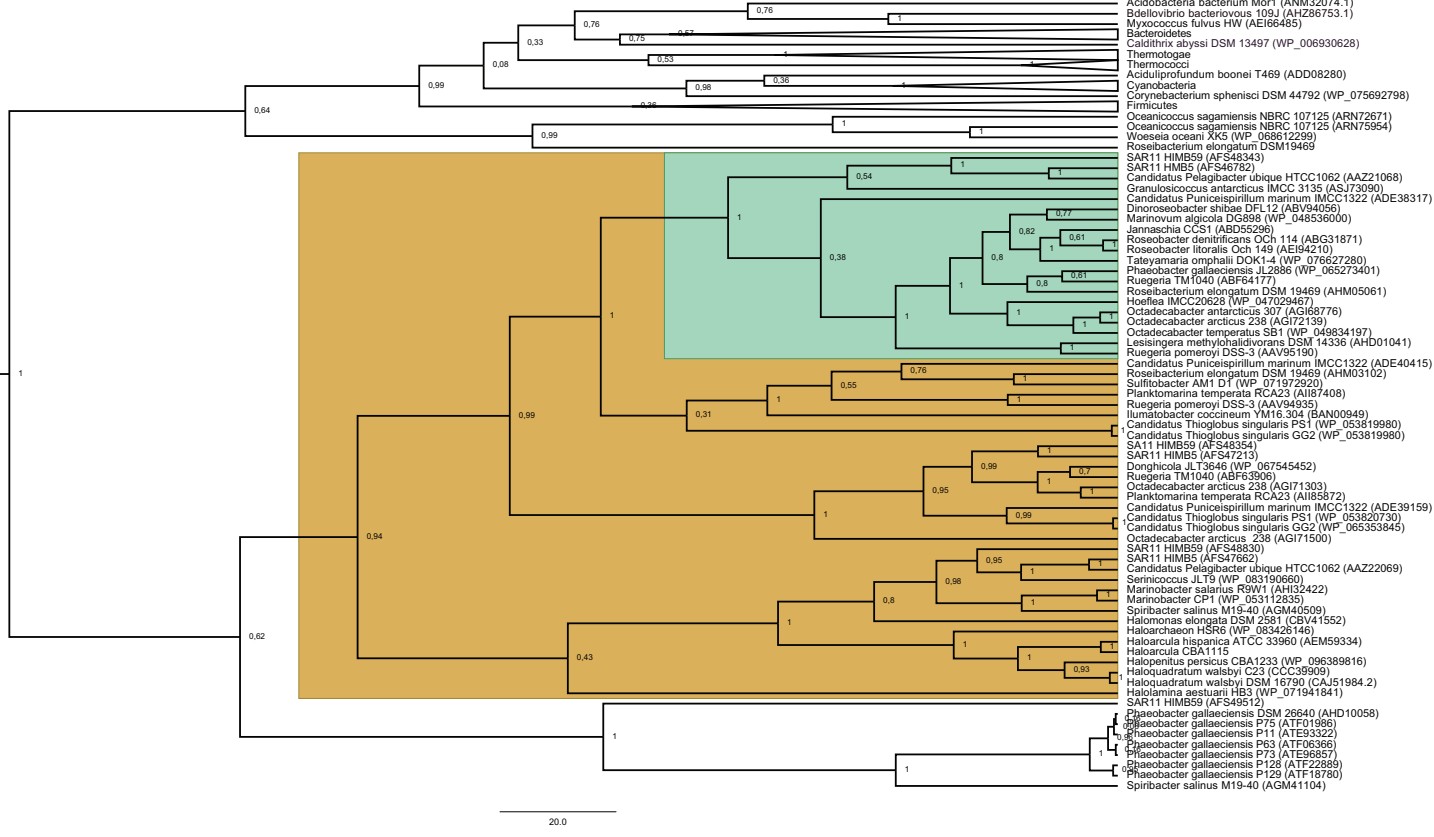

**Figure 1** Phylogenetic tree based on 20 DmdA orthologs protein sequences and 184 DmdA homologs using BEAST2 and the same parameters set for molecular dating but with 100 million generations. DmdA sequences are indicated with green color and closer homologs (the limit to select the closer homologs was set to a maximum *e*-value of E−80) with brown color. Tip labels include a maximum *e*-value of E−50.

different. Because the random split distance resulted in a value close to 1 (0.988), our observations are unlikely to be given by chance.

To identify HGT and duplication events, we constructed a proxy for the species tree of the genomes considered here by using a set of small subunit ribosomal protein (see Material and Methods). Given this (proxy) species tree (Fig. S6), the positions of many sequences on the DmdA tree are better explained as cases of HGT (Fig. S6; Fig. 3) with high statistical support. Then we tested whether the topology for a common set of taxa within the DmdA family (Fig. S7) similar to that of the species tree (Fig. S8). We found significant differences (at an alpha of 0.01) between the topology of DmdA group and that of the proxy species tree (Table S4); this incongruence between phylogenies is conserved irrespective of the test used (KH, SH and unbiased tests). From these results, we concluded that the phylogenetic relationships within each DmdA group were different to those of the species tree, strongly supporting a HGT-based evolution of DmdA family (Fig. S8). Moreover, we found many genes that use different codons from the neighboring genomic regions. These genes are inferred as having been horizontally transferred given their (G+C) wobble content (Table S1), supporting HGT as a plausible
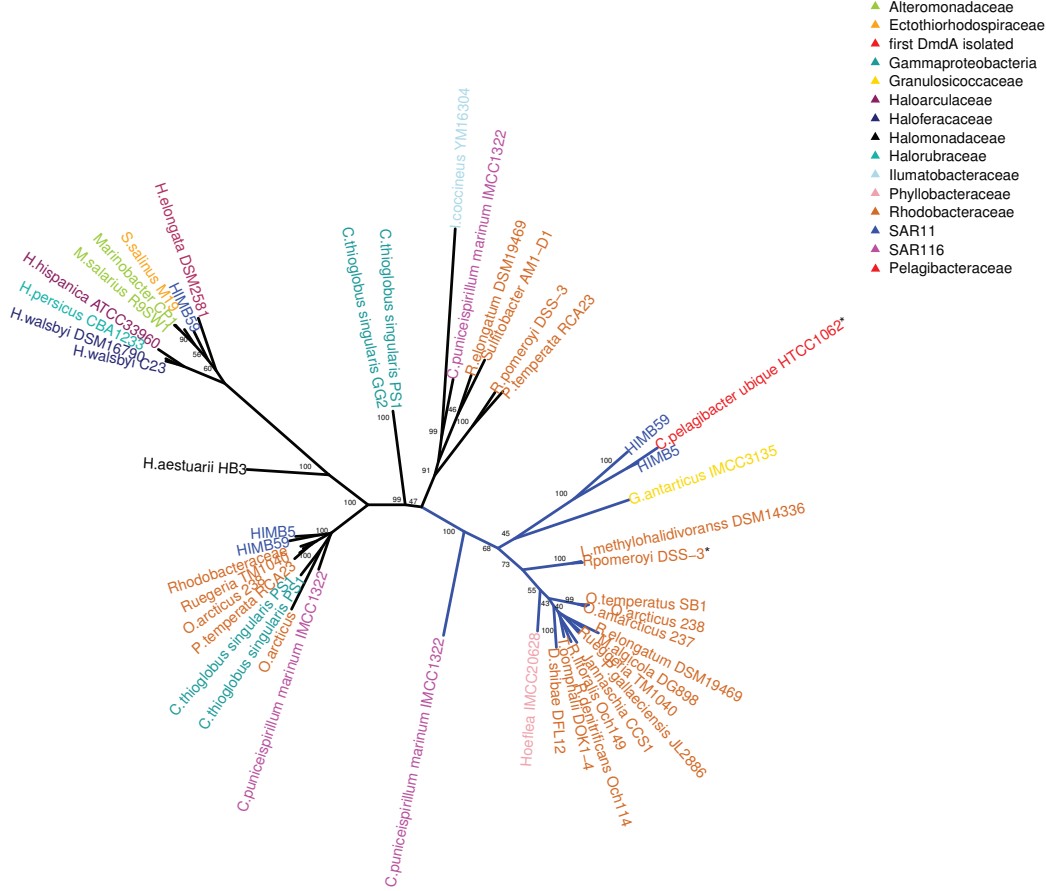

**Figure 2 RAxML phylogenetic tree built with 20 DmdA ortholog protein sequences and 28 DmdA homologs (more information in Table S1).** Non-parametric bootstrap values are shown to establish the support for the clades. DmdA sequences are indicated with blue branches. Tip labels show color according to their taxonomy classification and the asterisk indicates the first gene identified experimentally. Tip labels include a maximum *e*-value < E−80.

mechanism of genomic variability which introduces more variation than vertical gene transfer (VGT) and that contribute to DmdA evolution (Fig. S8).

## Structural modeling

The structure for DmdA orthologs inferred on the protein sequences by I-TASSER were threaded onto the known structure of DMSP-DmdA (PDB accession: 3tfhA) with a *C*-score <= 2 (Table S5). However, the predicted models for DmdA homologs were threaded onto two types of known structure; DmdA orthologs, and the structure of the mature form of rat dimethylglycine dehydrogenase (DmgdH) (PDB accession, 4ps9sA) with a *C*-score < 2 except for the sequence with accession number AEM59334.1, which showed a C-score > 2 (Figs. S9a and S9b; Data S1).

  We clustered sequences with a putative DmgdH structure in a separate group using principal component analysis (Fig. S9c). There is a clear 3D-structure coincidence between DmdA clade (green color in Fig. S9a) and the majority of lineages from non-DmdA clade (brown color in Fig. S9a), as well as a conserved folate-binding domain (Fig. S9b: 99S,

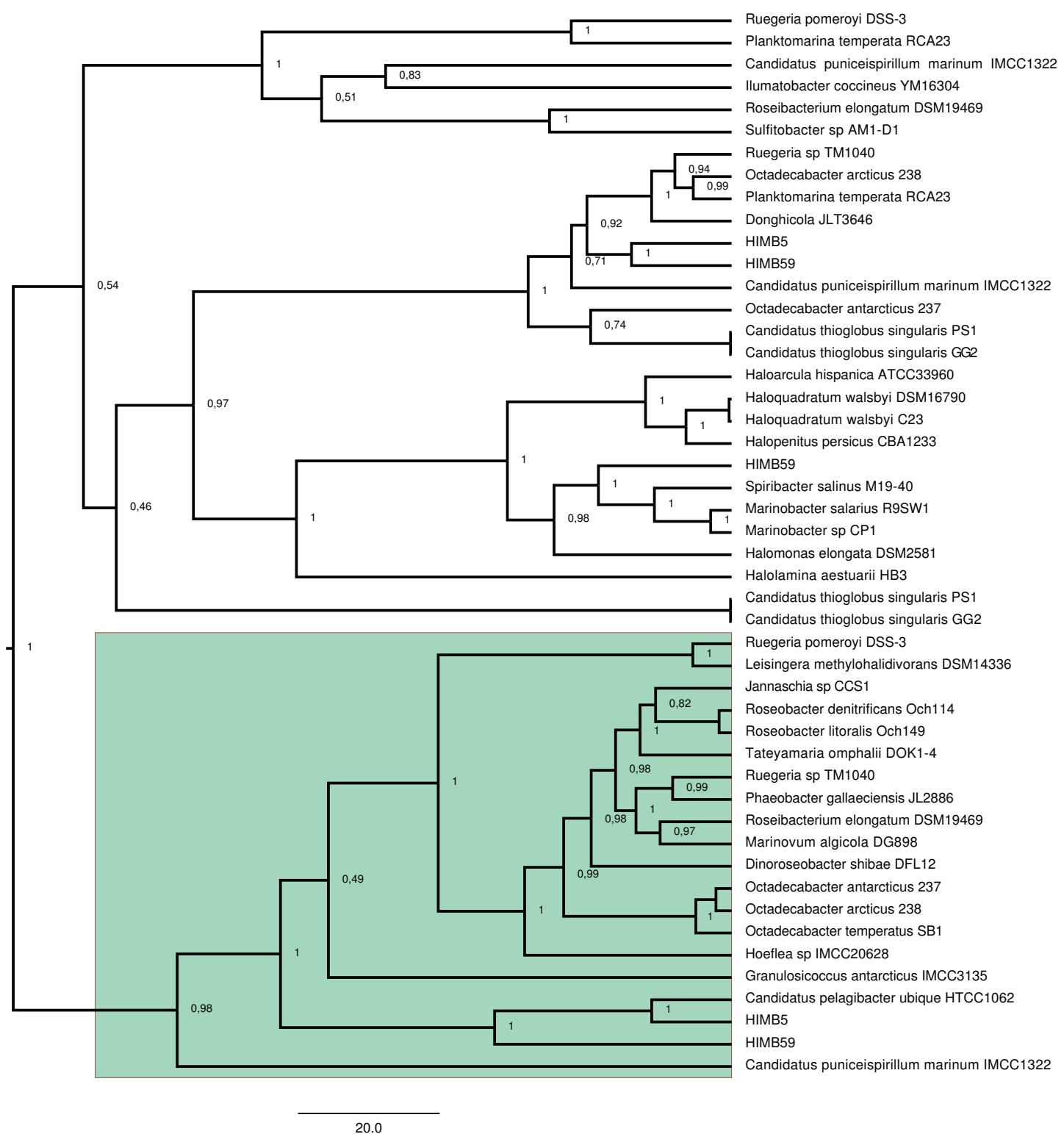

**Figure 3 BEAST2 phylogenetic tree constructed with alignment of 20 DmdA ortholog protein sequences and 28 DmdA homologs.** Bayesian posterior probabilities (PP) are shown to establish the support for the clades. Green color indicates DmdA clade.

178E and 180Y). However, in the alignment we found a pattern of conserved residues coherent with the phylogenetic results (Figs. S9a and S9b), where non-DmdA clade is formed by three subclades, one of them with DmgdH tertiary structure. Indeed, a key residue for DMSP specific interaction is shown in clades with DmdA tertiary structure (Fig. S9b: W171), but not in a clade with DmgdH tertiary structure (Fig. S9b: F171).

## Molecular dating

The log likelihood test (LRT) detected heterogeneity in the substitution rates of *dmdA* orthologs and *dmdA* homologs genes (Fig. 2) (log $L_0$ = −29,827.108; log $L_1$ = −29,546.053; degrees of freedom = 46; chisq = 562.11; $P$ < 0.001), thus rejecting the hypothesis of a strict molecular clock. This finding validates the use of a relaxed molecular clock approach to estimate the node ages through Bayesian analysis (see "Methods" for details).

We observed that the marginal densities for each run of the divergence time estimate analysis were nearly identical, pointing that the runs converged on the same stationary distributions. In all runs the marginal densities for the standard deviation hyperparameter of the uncorrelated log-normal relaxed clock model were quite different from the prior, with no significant density at zero, and with a coefficient of variation around 0.2. Analyses using three different calibrated prior dates showed no discrepancies in the final divergence time estimates (Table S6).

The time estimates for the MRCA of each gene family (Table S6; Fig. 4) indicate that the MRCA of DmdA gene family occurred in the late Archean, around 2,400 Mya, after a gene duplication event. Also, a duplication within the DmdA lineage generated a separated SAR11 and Roseobacter DmdA lineage in the early Precambrian ca. 1,894 Mya (Fig. 4: red arrow). *Ca.* P. ubique HTCC1062 within the SAR11 cluster and *R. pomeroyi* DSS-3 within the Roseobacter cluster, resulted from a duplication around 300 Mya (Fig. 4: blue arrow). However, a higher number of duplication events took place in the second cluster (Fig. 4: green color).

We detected two duplication events within the putative non-DmdA clade (Fig. 4; brown color); showing that the gene families were originated through old duplication events. One duplication involving the DmgdH family (Fig. 4: light yellow color; Table S5) occurred ca. 1,480 Mya and another duplication ca. 1,000 Mya (Fig. 4: green arrow), involving a gene family with tertiary structure similar to *Ca.* P. ubique DmdA The other duplication event took place during the Huronian glaciation, around 2100 Mya (Fig. 4: violet arrow).

## Reconstruction of ancestral DmdA sequence

Our analysis was focused on the reconstruction of the ancestral sequences of the DmdA clade, the non-DmdA clade as well as the ancestral sequence of both the DmdA and non-DmdA clades. FastML inferred the 100 most likely ancestral sequences of the DmdA family. We observed that the same sequences were always inferred. Indeed, the difference in log-likelihood between the most likely ancestral sequence at this node (N1; Fig. S10) and the 100th most likely sequence was only 0.105, indicating that both sequences were almost as likely to reflect the "true" ancestral sequence. That ancestral

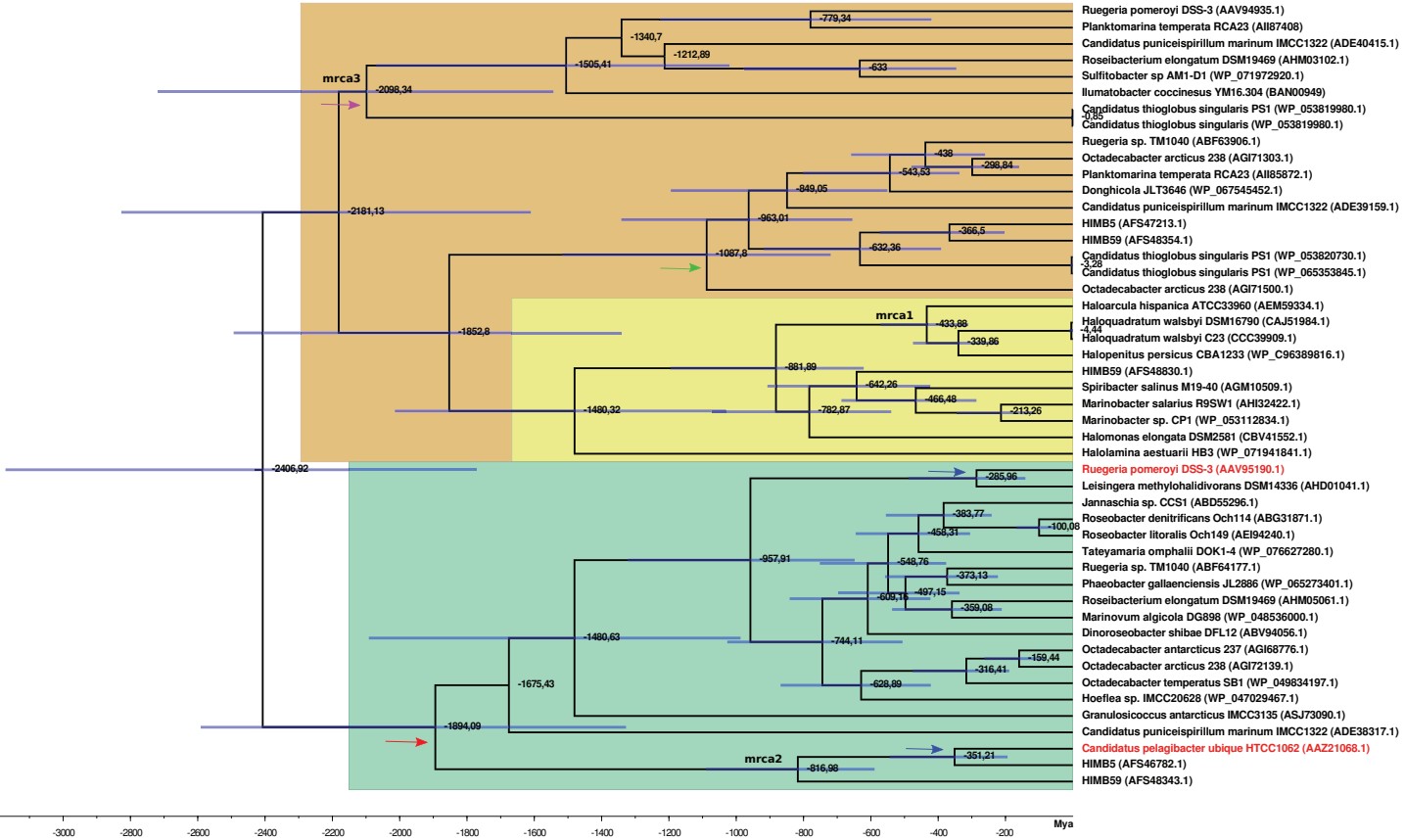

**Figure 4** **BEAST2 divergence time estimates from *dmdA* and non-*dmdA* genes under uncorrelated relaxed clock model and Birth-death tree model. Absolute time scale in Mya.** Nodes are at mean divergence times and blue bars represent 95% HPD of node age. Nodes used as calibrated priors in BEAST2 analysis are marked as mrca1, mrca2 and mrca3. Arrows indicate duplication events occurred 1,894 Mya (red), 300 Mya (blue), 1,000 Mya (green) and 2,100 Mya (violet). The predicted non-DmdA clade is shown in brown, DmgdH gene family is in light yellow and the DmdA clade in green color.

protein contains both PF01571 (GCV_T) and PF08669 (GCV_T_C) domains, found in the DmdA orthologs and it is nearly identical to *Ca*. P. ubique HTCC1062 DmdA sequence. Moreover, PSI-BLAST search confirmed that the ancestral sequence in node 1 close to DmdA genes hosted in EMBL-EBI databases (Fig. S11) and the structure for *Ca*. P. ubique apoenzyme DmdA was the closest analog to our predicted models (Table S5; Data S1). Inferred physico-chemical properties are identical between *Ca*. P. ubique and the DmdA ancestral sequence (Table S7).

On the other hand, the ancestral sequence inferred for non-DmdA family (N1; Fig. S12) and the ancestral sequence previous to functional divergence (N1; Fig. S13) contains only the PF01571 domain. That domain was located onto the known structure of T-protein of the Glycine Cleavage System (PDB accession: 1wooA) with a *C*-score= 1.25 (Table S5; Data S1) in the case of the ancestral DmdA and non-DmdA sequence. However, the ancestral sequence for non-DmdA was better threaded onto the known structure of mature form of rat DmgdH (PDB accession: 4p9sA) with a *C*-score= 0.76 (Table S5; Data S1).

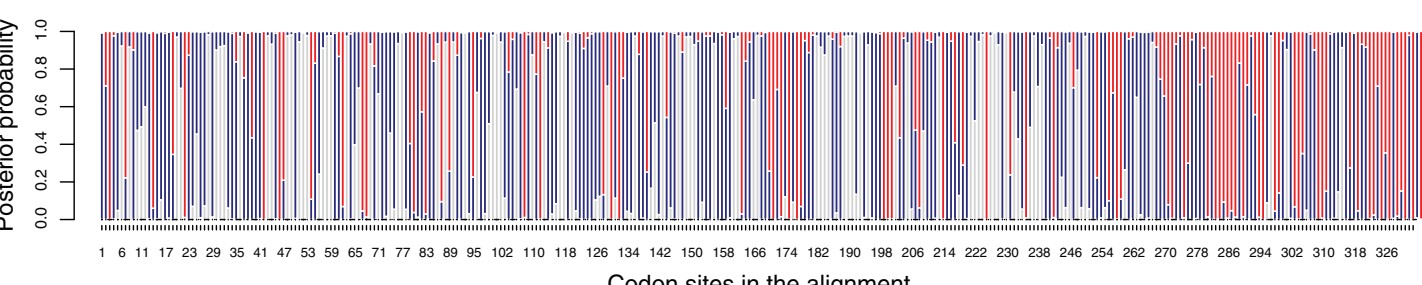

**Figure 5** **Posterior probabilities for dN/dS categories under the M3 model. Grey, red and blue bars depict the three dN/dS categories (values for each category are provide in the key).** Sites that are mostly grey denote codons under strong purifying selection, whereas those predominantly red show codons under weaker purifying selection. Red, blue and grey colors indicate codon sites with $\omega_2 = 0.2483$, $\omega_1 = 0.06923$ and $\omega_0 = 0.00485$, respectively.

### Detection of positive selection on *dmdA* sequences

To infer how natural selection has influenced the evolutionary history of DmdA gene family, we used an alignment of the 20 sequences clustered as *dmdA* orthologs (Fig. S14). The phylogenetic tree for these sequences was constructed by ML using the symmetrical model (SYM) with a discrete gamma distribution.

The average dN/dS value for the *dmdA* gene was 0.085, suggesting that this gene evolved under strong negative (purifying) selection. Then, we analyzed dN/dS variation across the codons in the gene, comparing M0 and M3 models through a LRT. The M3 model fits the data better than the M0 model (chisq = 775.387, *p*-value < 0.01). All codons in the gene are under strong purifying selection with dN/dS < 1 (Fig. 5), which indicates that this sulfur pathway is important for the cells. In accordance with this, the LTRs designed to detect codons under positive selection were not significant (M1 vs. M2, chisq = 0 and *p*-value = 1, and M7 vs. M8, chisq = 1.459 and *p*-value = 0.482). Hence, we did not detect sites in *dmdA* subjected to positive selection (Fig. S15).

We tested the variation in the intensity of selection over evolutionary time. A two-ratio model comparing the Roseobacter with the rest of lineages (Fig. S16) fits the data, as the LRT was 23.777 and *p*-value < 0.01 (Table S8). dN/dS value in Roseobacter ($\omega_1$: 0.0767) was significantly lower than in the remaining branches ($\omega_2$: 0.1494), suggesting stronger purifying selection on *dmdA* in Roseobacter. When we tested the intensity of selection over evolutionary time using the free-ratio model (Table S8), we found changes in the selection pressure from the branches which defines the separation of SAR11 from Roseobacter DmdA gene families (Fig. S17: branches from nodes 21 to 23). In particular, we observed a dN/dS value > 1 in the branch connecting nodes 21–23. We also identified some more recent branches (connecting nodes 25–26 and 28–29) for which dN/dS >> 1 was estimated (Fig. S17).

Finally, we applied the two branch-site models to test for sites under selection on the individual lineages associated with *dmdA* (Fig. S18). Four sequences (WP_047029467, AHM05061.1, ABV94056.1, AFS48343.1) had a significant LRT after correcting for

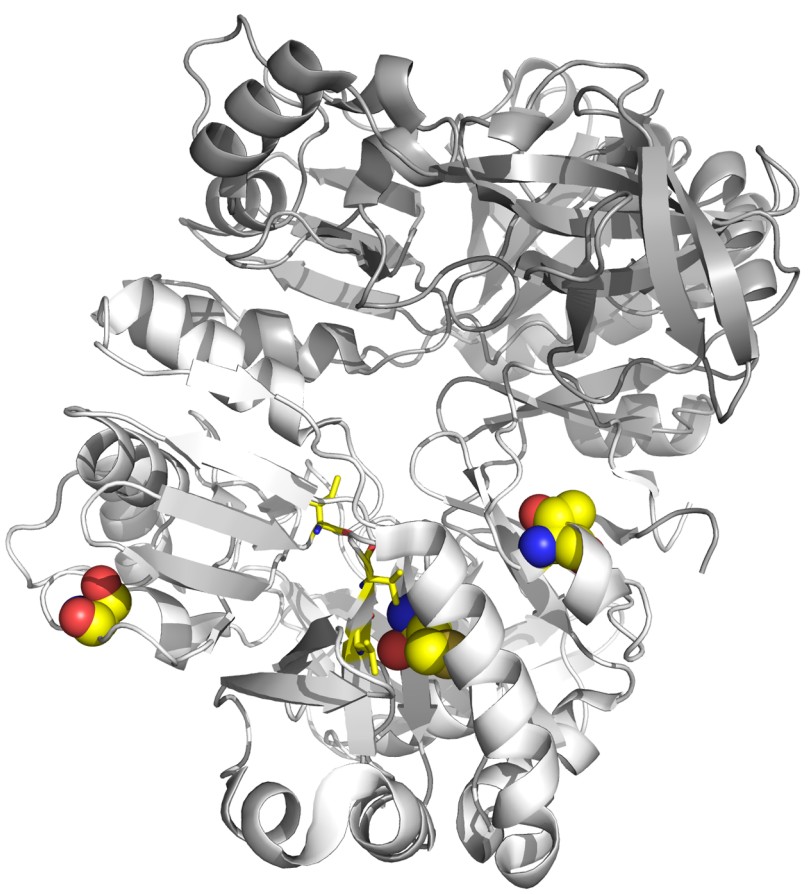

**Figure 6 Tertiary structure of DmdA (PBD: 3tfh belongs to Ca. P. ubique) with sites under episodic positive selection.** Sites under positive selection are shown in chain B (white color) as spheres with oxygen in red, nitrogen in blue, hydrogen in white and sulfur in yellow. THF binding sites are shown as yellow sticks.

multiple testing (Table S9), corresponding to episodic positive selection on these lineages (Fig. S18). It should be highlighted that three selected sites are shared by at least two lineages (Table S9; Fig. 6). One shared site is located next to the GcvT domain (152 K; Fig. S19), and two shared sites are close to conserved positions (17E; 87Y; Fig. S19). The residue 87Y is adjacent to the conserved interaction site with THF (88Y; Fig. S19). Interestingly, since the selected lineages are separated in the tree, the adaptive mutations seem to have occurred through three parallel independent changes (Fig. S20).

## Functional divergence during the molecular evolution of DmdA sequences

We tested whether DmdA and non-DmdA gene families were subject to different functional constrains after gene duplication (Fig. S5). We estimated the one-ratio model (M0) that yielded a value ω = 0.053 (Table S10), indicating that purifying selection dominated the evolution of these proteins. The discrete model (M3) was applied to these sequences (Table S10) and the LRTs comparing M0 and M3 indicated significant variation in selective pressure among sites (Table S10; Fig. S21).

The M3 model was compared with Model D, which accommodates both heterogeneity among sites and divergent selective pressures. The LRT was significant and supported the model D (Table S10), implying statistical evidence for functional divergence between DmdA and non-DmdA. Parameter estimates under Model D with $k = 3$ site classes suggested that 23.6% of sites were evolving under strong purifying selection ($\omega = 0.006$), while 26.7% of sites were evolving under weaker selective pressure ($\omega = 0.04$). Interestingly, a large set of sites (49.6%) were evolving under divergent selective pressures, with weaker purifying selection in the DmdA-clade ($\omega = 0.169$) than non-DmdA-clade ($\omega = 0.100$). We identified 77 sites evolving under divergent selective pressures between DmdA and non-DmdA (Table S10). Nineteen sites were located within the alpha helix (red tube in Fig. S22) of the secondary structure prediction and sixteen were located in the beta sheet (green arrows in Fig. S22). According to the global dN/dS estimates, for all divergent positions, *dmdA* sequences seem to be more conserved than non-*dmdA* sequences. Moreover, this data were only compatible with recombination breaking linkage disequilibrium within the gene set that we observed with the HGT analysis.

Finally, we were interested in finding out if adaptive evolution has occurred in the lineages immediately following the main duplication event (Fig. S23). We applied two branch-site models to test for sites under selection on the ancestor associated with the DmdA and non-DmdA clades (Table S9). The LRT was significant for both ancestral branches (LRT > 7 and *p*-value < 0.05). Nonetheless, the foreground $\omega$ for class 2 sites tended to infinity ($\omega = 999$) in both cases, indicating lack of synonymous substitutions (dS = 0) in these sites. We also performed two-ratio models to estimate global $\omega$ on these branches, but both estimates tended to infinity (Table S11), suggesting lack of synonymous substitution in the divergence of DmdA and non-DmdA ancestors. Therefore, although the fixation of only non-synonymous substitutions following gene duplication might indicate strong positive selection driving functional divergence of DmdA and non-DmdA families, we cannot confirm it with the applied tests.

## DISCUSSION

In this study we evaluated three scenarios for the evolutionary history of the DmdA gene family in marine bacteria. The results for each one are discussed separately.

### First scenario: a recent common ancestry between DmdA and GcvT

In relation to the first scenario, we found that contrary to our initial expectations, DmdA and GcvT do not seem to have a recent common ancestry, in constrast to DmdA and non-DmdA. The clear separation between DmdA and putative non-DmdA gene families that originated in the Archean ca. 2,400 Mya after a gene duplication, supports a common recent ancestry for DmdA (Fig. 7A) and non-DmdA (Fig. 7B). Our tertiary structure analyses indicate that they share a putative GcvT protein as their ancestor sequence (EC 2.1.2.10). Indeed, our results agree with other studies in the case of DmdA (*Reisch, Moran & Whitman, 2008*). Then, this clade seems to have been originally a GcvT (Fig. 7), as *Bullock, Luo & Whitman (2017)* suggested.

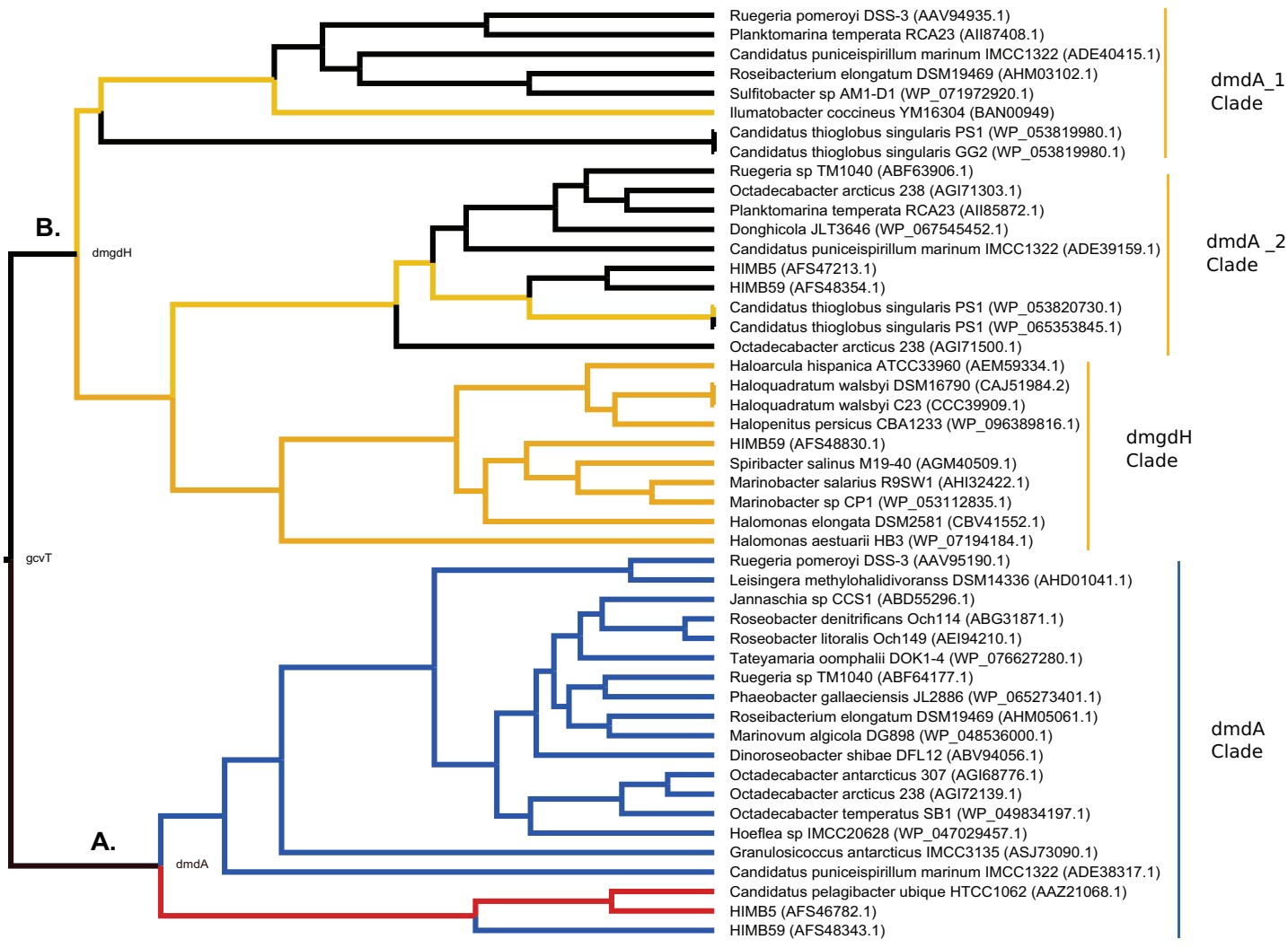

**Figure 7 Hypothesis of DmdA evolution. BI phylogeny under uncorrelated relaxed clock model and Birth-death tree model.** Node names represent the reconstructed ancestral sequences; GcvT family before to main duplication, DmdA for DmdA clade (A) and DmgdH for non-DmdA clade (B). In DmdA clade (A), blue color represents eco-orthologs where pI (predicted isoelectric point value) is < 5.7 and they are adapted to low concentration of DMSP in comparison with DmdA orthologs (red color) which have pI => 6.5. In non-DmdA clade (B), yellow branches represent homologs with DmgdH tertiary structure and black branches homologs with DmdA tertiary structure.

The DmdA clade is a member of aminomethyltransferase (AMT/GCV_T) family with DMSP-dependent demethylase tertiary structure, while non-DmdA clade includes an ancestor with a tertiary structure that better matches the dimethylglycine dehydrogenase oxidorreductase (DmgdH, EC 1.5.99.2) (Fig. 7B) and members with DmdA tertiary structure. To establish structural convergence as the reason of this DmdA structure coincidence between DmdA and non-DmdA members, we used a phylogenetic approach based on reconstructing ancestral sequences of the two clades, and then to model the ancestral proteins. We determined different structural features between ancestral sequence reconstructed from DmdA and non-DmdA families. In the first case, the ancestral sequence reconstructed coincides with a DmdA tertiary structure (Fig. 7A), as well as with a DmdA sequence with physico-chemical properties inferred in this study (Table S7) and

agree with previous ones (*Reisch, Moran & Whitman, 2008*). However, the non-DmdA ancestral sequence reconstructed is a DmgdH that seems to be kept in the clade called DmgdH (Fig. 7B), as well as in some members of DmdA clades (DmdA_1 and DmdA_2 within non-DmdA clade) where the majority of sequence gained DmdA structure (Fig. 7B). Therefore, DmdA structural features seem to have emerged independently in both clades: DmdA and non-DmdA. This finding is interesting, since known cases of structural convergence of proteins are rare (*Zakon, 2002*). Experimental assays expressing and screening the activity of the ancestral proteins at different conditions will be required to corroborate the structural convergence.

Since GcvT does not share the most recent common ancestry with DmdA (as we observe in Fig. 7), we examined the functional divergence between DmdA and non-DmdA clades to explain how natural selection could have driven the divergence of the DmdA gene family. We found 77 codon sites evolving under divergent selective pressures between DmdA and non-DmdA gene families. Structural divergence seemed to be imposed on the protein during sequence divergence, since 19 sites were located within the alpha helix of 2D structure and 16 in the beta sheet (Fig. S22). Nonetheless, essential regions of the enzymes as active sites seem to be under strong purifying selection, suggesting preservation of the ancestral function. The observation that DmdA sequences have more divergent sites than non-DmdA sequences suggest that non-DmdA conserves the ancestral function, whereas DmdA evolved to acquire new functions in different environments, probably as a response to the Huronia ice ball Earth (*Zhang, 2003*).

## Second scenario: coevolution between Roseobacter and DMSP-producing-phytoplankton

In the second scenario, our results do not support the hypothesis of a co-evolution scenario between Roseobacter and DMSP-producing-phytoplankton (*Luo et al., 2013*). On the contrary, we found an ancestor sequence of DmdA cluster similar to DmdA from a strain of *Ca*. P. ubique that diverged after a more recent duplication event (Fig. 7A; Fig. S10), before the dinoflagellate radiation in the late Permian (Fig. 4). This finding indicates that the enzyme activity has not changed in the course of DmdA evolution and is in Roseobacter because their genome expansion (250 mya) provided a new trait to use DMSP produced by phytoplankton during its diversification. Indeed, we found that most of the codons in DmdA clade are under purifying selection, probably due to the importance of this pathway for sulfur acquisition. Nonetheless, we also detected episodic positive selection in four sequences affecting a few sites, suggesting that adaptive evolution fine-tuned the function of DmdA in Roseobacter and other types of *Alphaproteobacteria* (like HIMB59 and *Hoeflea*). Furthermore, positively selected residues were located around the GcvT domain and close to the residue involved in conserved interaction with THF (Fig. 6), reinforcing the idea of adaptive evolution in response to the external environment.

During the study of this scenario, we suspected that *dmdA* was acquired by HGT in Roseobacter and SAR11 (Fig. S8). This agrees with *Luo et al. (2013)* and *Tang et al. (2010)* which found that the expansion of *dmdA* resulted from HGT events. According to our phylogeny, the ancestral *dmdA* sequence originated as a result of HGT (in individuals not

connected by inheritance that acquired the *dmdA* ancestral sequence) from other marine heterotrophic bacteria, that during the Archean adapted to the presence of DMSP. However, after the HGT events, some *dmdA* sequences have acquired similar residue changes by independent (parallel) evolution, reinforcing the idea of functional/ecological constrains (*Siltberg-Liberies, Grahnen & Liberies, 2011*). Therefore, *Rhodobacteraceae* can live in an environment where DMSP is the main source of sulfur because they acquired the *dmdA* ancestor sequence by HGT, prior to having been exposed to the environment in which the DmdA protein proved useful, as *Luo & Moran (2014)* suggested. We did not find any signal of positive selection in the Roseobacter group, but in contrast we found episodic evolution between SAR11 sequences. Yet, as we already mentioned, DMSP is part of an ancient pathway in *Alphaproteobacteria* (*Bullock, Luo & Whitman, 2017*) and this could explain the ancient origin of DmdA.

On the other hand, Roseobacter orthologs analyzed in this study were functionally annotated as DmdA (*González et al., 2019*), as they were predicted to originate from the same DmdA ancestor. However, we identified orthologs within DmdA gene family as *Sánchez-Pérez et al. (2008)* proposed in their study regarding related genes that perform the same cellular function, but apparently under different ecological conditions, as we found differences in predicted isoelectric point values (pI) (Table S7). *Nandi et al. (2005)* results also support that orthologs with very variable pI values may be taken as markers to predict the organism's ecological niche. We suggest the name "eco-orthologs", similar to the ecoparalogs describe by *Sánchez-Pérez et al. (2008)* in their study of the halophilic species *Salinibacter ruber*. The pI values of a protein provide an indication of its acidic nature on the surface, corresponding to its optimal activity and stability at high salinity (*Oren et al., 2005*; *Sánchez-Pérez et al., 2008*). Therefore, proteins that differ in their acid residue content on their surface, and consequently in their predicted pI values and halophilicities are considered eco-orthologs (*Nandi et al., 2005*; *Oren et al., 2005*; *Sánchez-Pérez et al., 2008*). We observed the highest pI values in the DmdA ancestor sequences, as well as in *Ca*. P. ubique DmdA (Fig. 7A; red color). Therefore, we deduce that the DmdA ancestor was adapted to a higher salinity, which could have modulated the selection of the DMSP enzymatic degradation routes as in bacteria such as the model organism *R. pomeroyi* DSS-3 (*Salgado et al., 2014*). Interestingly, *R. pomeroyi* degrades more DMSP by the demethylation pathway under high salinity conditions, releasing a higher amount of MeSH (*Howard et al., 2008*; *Magalhães et al., 2012*; *Salgado et al., 2014*). The success of the *dmdA* gene could be explained if we consider that the environment evolved from higher to lower salinity conditions. Under this environment, *dmdA* would have been kept without important changes in its structure, sequence, function and $K_m$ value and even would be essential for the degradation of the large amounts of DMSP produced by phytoplankton. Indeed, it would be interesting to evaluate $K_m$ values among ancestral proteins of DmdA and their descendants to support the key role of $K_m$ during DmdA evolution. In addition, since *dmdA* seems to be part of a conserved operon (*González et al., 2019*), its evolution might be linked to genes such as *dmdB*, *dmdC* and *dmdD* that encode part of the enzymes for the rest of the pathway.

Given our data, we propose that the ancestor of the pathway that evolved during the Archean was exposed to a higher concentration of DMSP in a sulfur-rich atmosphere and in an anoxic ocean, compared to recent eco-ortologs which should adapt to lower concentration of DMSP (Fig. 7A: blue color). Indeed, the ancestral eco-orthologs from which recent eco-orthologs derived (*Candidatus* Puniceispirilum marinum IMCC1322, ADE38317.1 and the Roseobacter clade) could have undergone episodes of adaptation (the branch showed positive selection in branch-models) which would explain the change in protein stability (*Pál, Papp & Lercher, 2006*). As consequence, the protein could have experienced slight reductions or loss of function.

### Third scenario: pre-adapted enzymes to DMSP prior to Roseobacter origin

In this evolutionary scenario, the Roseobacter clade was pre-adapted to the conditions created by eukaryotic phytoplankton of the late Permian, including dinoflagellates that released vast amounts of DMSP (*Bullock, Luo & Whitman, 2017*; *Luo & Moran, 2014*). Our analyses indicate that the Roseobacter ancestor was already adapted to a high DMSP before the Roseobacter clade arose (*Luo et al., 2013*). Therefore, we support *Reisch, Moran & Whitman (2011)* and *Reisch et al. (2011)* hypothesis that DMSP demethylation pathway enzymes are an adapted versions of enzymes that were already in bacterial genomes and that evolved in response to the availability of DMSP. Since the first step in DMSP demethylation is a reaction catalyzed by DMSP demethylase encoded by *dmdA* gene (*Dickschat, Rabe & Citron, 2015*), DMSP adaptation could have been evolved in this gene that originated in the Archean, a time where several lineages of bacteria produced DMSP as an osmolyte or antioxidant in the presence of the early cyanobacteria, or as a cryoprotectant in the Huronian glaciation. In bacteria, a methyltransferase gene, *dysB*, is up-regulated during increased salinity, nitrogen limitation, and at low temperatures (*Curson et al., 2017*), conditions already predicted to stimulate DMSP production in phytoplankton and algae (*Bullock, Luo & Whitman, 2017*; *Ito et al., 2011*). Afterward, those roles may have helped to drive the fine adaptation of existing enzymes for DMSP metabolism, and those adaptations came handy in the late Precambrian glaciations that allowed the radiation of algae and animals.

## CONCLUSIONS

In conclusion, we found that Roseobacter adaptation to DMSP occurred via functional diversification after duplication events of the *dmdA* gene and adaptations to environmental variations via eco-orthologs of intermediate divergence. Our findings suggest that the DmdA ancestor evolved to play a key role in the ocean sulfur cycle due to a shift in salinity concentration, which involved a change in DMSP synthesis.

## ACKNOWLEDGEMENTS

We would like to thank to Dr. Romain Studer from BenevolentAI for his critical role in the 3D visualization of protein and mapping sites onto the 3D structure and Dr. Buckley

Iglesias from Universidad Autonóma de Madrid for his introduction to molecular dating analysis with BEAST 2.

### Funding
This research was supported by grant CTM2016-80095-C2 from the Spanish Ministry of Economy and Competitiveness. The funders had no role in study design, data collection and analysis, decision to publish, or preparation of the manuscript.

### Grant Disclosures
The following grant information was disclosed by the authors:
Spanish Ministry of Economy and Competitiveness: CTM2016-80095-C2.

### Competing Interests
Luis Enrique Eguiarte and Valeria Souza are Academic Editors for PeerJ.

### Author Contributions
- Laura Hernández conceived and designed the experiments, performed the experiments, analyzed the data, prepared figures and/or tables, authored or reviewed drafts of the paper, and approved the final draft.
- Alberto Vicens conceived and designed the experiments, performed the experiments, analyzed the data, prepared figures and/or tables, authored or reviewed drafts of the paper, and approved the final draft.
- Luis E. Eguiarte conceived and designed the experiments, analyzed the data, authored or reviewed drafts of the paper, and approved the final draft.
- Valeria Souza analyzed the data, authored or reviewed drafts of the paper, and approved the final draft.
- Valerie De Anda analyzed the data, authored or reviewed drafts of the paper, and approved the final draft.
- José M. González conceived and designed the experiments, authored or reviewed drafts of the paper, and approved the final draft.

### Data Availability
The data collected from MarRef database (including information about sequences and genomes used in this study, taxonomy and sampling environment) are available in Table S1.

### Supplemental Information
Supplemental information for this article can be found online at http://dx.doi.org/10.7717/peerj.9861#supplemental-information.

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
