# Peer review of "Evolutionary history of dimethylsulfoniopropionate (DMSP) demethylation enzyme DmdA in marine bacteria"

_PeerJ, doi:10.7717/peerj.9861_

## Round 0.1 · original submission · Minor Revisions

While both reviewers found general interest and promise in the topic of study, they have highlighted several areas in need of revision.

Reviewer 1 ·

Basic reporting

Text mostly well presented, though some confusions with the tense. The Tables and Figures could be better explained with improved and more substantive legends

Experimental design

My expertise is not suited to make accurate judgments on the approaches taken.

Validity of the findings

Fine as far as I can see. This is, however, has a strong theoretical flavour and so, even if their conclusions are not 100% correct, they recognise this and in any event, this work will form the basis for interesting discussions in the field.

Additional comments

Comments on PeerJ MS on DmdA - see also annotated Pdf

On reading this MS, I feel a little like the chap who wanders into the Prado muttering “I don’t know much about art, but I know what I like”. So, the good news is that I liked this paper, with its full-on bioinformatic attack on a single enzyme/gene system. But the bad news is that my abilities to make meaningful judgments on the fine details of these bioinformatic analyses would make Donald Trump blush. So, with that in mind, here goes.
Most of my comments (167 of them) can be seen in situ in the attached Pdf. Many are minor – the English and written presentation are mostly very good, but there are some glitches and these are indicated. One more important problem though, is the need to decide if they are writing in the present or past tense. They start in the former, then switch. There are also some more substantive comments (prefaced with a *** in the Pdf) and these need attention.
Here, I indicate my wider concerns.
I think that my overall worries relate to the fact that the DmdA/dmdA polypeptide/gene (by the way, there are several places where the nomenclature for these is mixed up and wrong) are considered in isolation, addressing them purely in terms of their sequences and their known and proposed structures. Given the terms of reference that they set themselves, that may be fair enough, but, no gene is an island, so maybe they need to bring in other factors.
So, a few things. Of the elephants in the room, the one that gives me most cause for concern is the lack of any mention (I checked) of the worryingly (for some) high Km (~10 mM) values of those DmdA enzymes that have been studied. If the family has had such a long time to evolve, this almost implies that a high Km is an adaptive advantage, perhaps in the context of the high intracellular DMSP concentrations. I think that this point needs to be addressed somewhere. [And, yes, I know that some (though not all) of the Ddd lyases are even less efficient!]
Although they certainly do mention the recent observations that much (maybe most?) of the world’s DMSP is bacterial in origin, I actually feel that this may be a game changer in this field, especially in light of the findings by Williams et al (https://www.nature.com/articles/s41564-019-0527-1) of the range of different bacteria and different pathways that can accomplish this synthesis. This finding may not substantially change the overall line of thinking and the conclusions of this MS, but I do think that it changes the backdrop to the DMSP story and needs to be reflected. (I assure you that this is not a case of “point-scoring”.)
I would like to have seen some consideration of what they think are the features of the Roseobacter lineage (not Roseobacter in italics as repeatedly written in the MS) that makes them so DMSP-philic. If the dmdA functional gene has been around for such a long time, why did it not find its way into a much wider range of marine bacteria? After all, there are very close homologues of the downstream DmdB, C and D polypeptides in a whole host of other bacteria, so these would only require the acquisition of the single dmdA gene for a functional pathway? And that is not all. Emphasising the importance of DMSP in the life and times of the Roseobacters, these bacteria are home to many of the Ddd lyases, and there are links between the two pathways, most obviously that in several Roseobacters, the expression of the dmdA is under the control of the key product (acrylate) of the lyase mediated cleavage. I am not sure what this means in relation to the evolution of the DmdA enzyme, but I suspect that it means something.
I thought that the figure legends could be generally improved, with more information and written more clearly. See my in situ comments on Fig 1 line 329 for example.
Overall, though, I think that this paper will be a useful addition to the DMSP literature and at the very least will set off some productive and interesting discussions.

Annotated reviews are not available for download in order to protect the identity of reviewers who chose to remain anonymous.

Reviewer 2 ·

Basic reporting

Comments on clarity/english in text:

Line 68-69: Consider rewording to make this sentence more accurate to your point, such as: “Compared to genes in the DMS-releasing pathways, dmdA is more frequently found in the genomes of oceanic bacteria.”

Lns 360-363: This doesn’t need to be a separate paragraph. You could add it onto the previous paragraph.

Lns 405-407: I would suggest changing the terms ‘first cluster’ and ‘second cluster’ since these are hard to interpret for finding things where things are on the tree.

Ln528 : What does Fig. 7; “down and up” mean?

Ln 525-528: contradictory phrases “have not a recent common ancestry” and “supports a common recent ancestry” in same paragraph.

Ln 568-569: Are you referring to Figure 4? It would be helpful here and throughout the Discussion to reference the results to which you refer.

Ln 581-584: I cannot understand this sentence.

Ln 600-602: please consider rewording “as well as Ca. P. ubique sequence and this last one has a pI similar to the first” I don’t know what first and last one means.

Ln 605: “degradates” to “degrades”

Ln 616: “experimented” to “experienced”

Ln 444 should say “…selection has influenced the evolutionary history…”

Ln 460 should say “…fits the data better…”

Ln 587 should day “…prior to having been…”


Comments on Figures, Tables, and Supplemental Material:

All Tables (1-6) can be moved to supplemental material.

Supplemental Table 1: I do not understand how to interpret the G+C wobble values, some cells have multiple values, some have decimals and others have commas. Please revise.

Legends for the Supplemental figures and tables are only found on the PeerJ website; final versions of the supplemental materials need to have these legends within the article documents.

Figure 1: In the legend, please define what is meant by ‘closer homologs’ for those with yellow color.

I’m not convinced that Supplemental Figure 6 is necessary. It’s hard to directly compare with Figure 3 because you’re using a different set/number of sequences. You then go on to do the same analysis/comparison with a smaller subset of sequences in Supplemental Figures 7-9 that accomplish your point about HGT.

Legends for the supplemental figures are sometimes too vague. For example, Supplemental Figures 8 and 9 have the exact same title.

There is a fair amount of color switching between figures. DmdA sequences are labeled in blue, red, green, depending on which tree it is – and there are many trees. This leads to reader fatigue. I would suggest trying to make your coloring schemes as consistent as possible.

Supplemental Figure 10b is divided into 4 clades which are supposed to correspond to Figure 4 and Supplemental Figure 10a, but I only see 3 clades. It would also help to include taxonomic labels on Supp. Fig 10b so it’s easier to compare between it and the trees.

Supplemental Figure 10b: what’s the difference between darker and lighter blue?

Supplemental Figure 11: All I see is a black box with squares and a yellow “L” line. Should there be labels somewhere? Are the points corresponding to the red and yellow sequences from 10a? Again, keeping consistent color coding would be really helpful.

Figure 4: Please offset the node numbers and bars so that it’s easier to read. Move the node numbers off of the node labels. Also provide a label for the time scale and increase the font size of the scale numbers. Define violet arrow in legend.

Supplemental Figure 20: I don’t think this figure is necessary or provides any information beyond the text. What’s written in the legend is all the info that is needed.

Figure 6: what do the colors other than blue represent?

Figure 7: define "pI"

Overall comment: My list of figure corrections is probably not exhaustive, I would suggest going back through everything to find other missing information, mismatched colors, labels that need definitions, etc.

Experimental design

See comments on conclusions that may relate to experimental design.

Validity of the findings

Comments on interpretation/conclusions:

I admit that I am only partly familiar with some of the techniques used in the manuscript. An over-arching question that I have is, How are your results/interpretations influenced by the fact that you are comparing DmdA vs a potentially “mixed bag” of other proteins (i.e. DmgdH + DmdA-like paralogs) that 1) may have multiple different functions and 2) represent different taxonomic groups?
For example, how is your comparison of purifying selection in DmdA versus non-DmdA clades (Lns 491-504) influenced by analysis of so many different taxa/potential functions?

Lns 407-408: If I understand correctly, by ‘second cluster’ this is the one in green (please clarify this in the text). You conclude that the number of paralogous genes is greater for Roseobacters versus SAR11. Does having greater representation of Roseobacter sequences on the tree influence this? What about the greater phylogenetic distance among members in the Roseobacters? I may be confused because I was under the impression that the green block included DmdA orthologs only.

Ln 546-548: I’m not sure I understand/agree with this conclusion. The authors say that this phenomenon would be rare, but could your interpretation potentially be wrong?
Firstly, is it true that both the dmdA_1 and dmdA_2 clades on the tree are all true DmdA sequences? (You reference a paper, Gonzalez, 2019, but this is based purely on computational analysis, not experimental verification of function) which gives me concern. For the tertiary structure analysis, you are feeding in known structures of DmdA and DmgdH, but you don’t have an accurate structure of whatever makes up the dmdA_1 and dmdA_2 clades. Could it be that you are seeing an effect of not having a good predicted model for dmdA_1/2, and those sequences are just matching up with the best hit, which is the "true" DmdA model? This goes back to my concerns about interpreting the results when you have these ‘mixed bags’ of protein sequences that could represent multiple, different functions. Alternatively, what if the ancestor of the protein famil looked more like DmdA; you are only two yellow nodes away from a different conclusion (Fig 7)?

Additional comments

The authors use a suite of phylogenetic and computational approaches to reconstruct the evolutionary timeline and history of DmdA in marine bacteria. The topic is timely and interesting overall, and there has been recent literature that questions when/how DMSP degradation genes evolved (with some controversy). So, I applaud the authors for tackling this question. My main issue with the manuscript has to do with clarity. Many figures need better descriptions/definitions in the legends; at times, it was challenging as the reader to interpret results because of this. There are also many figures and tables, and the paper would benefit from more consistency in colors/labels and possibly removing/consolidating some figures. I also question some of the conclusions that the authors draw, but these concerns may be addressed through clearer language and explanation in the text. My specific comments are listed herein.

---

## Round 0.2 · accepted · Accept

Thank you for the thorough and thoughtful responses to the reviews, and congratulations on the acceptance of your manuscript. I read it and enjoyed it very much - it is a valuable contribution to the discipline.